# Influence of Photosensitizer on Photodynamic Therapy for Peri-Implantitis: A Systematic Review

**DOI:** 10.3390/pharmaceutics16030307

**Published:** 2024-02-22

**Authors:** Thaís B. M. O. Schweigert, João P. R. Afonso, Renata K. da Palma, Iransé Oliveira-Silva, Carlos H. M. Silva, Elias Ilias Jirjos, Wilson Rodrigues Freitas Júnior, Giuseppe Insalaco, Orlando A. Guedes, Luís V. F. Oliveira

**Affiliations:** 1Health Sciences, Post-Graduation Program, Faculty of Medical Sciences of Santa Casa de São Paulo (FCMSCSP), São Paulo 01224-001, SP, Brazil; thaisbuenomom@hotmail.com (T.B.M.O.S.); elias.ilias@fcmsantacasasp.edu.br (E.I.J.); wilsonrfreitasjunior@gmail.com (W.R.F.J.); 2Human Movement and Rehabilitation, Post-Graduation Program, Evangelical University of Goiás—UniEVANGÉLICA, Anápolis 75083-515, GO, Brazil; joaopedro180599@gmail.com (J.P.R.A.); rekellyp@hotmail.com (R.K.d.P.); iranse.silva@unievangelica.edu.br (I.O.-S.); carloshmendes@unievangelica.edu.br (C.H.M.S.); 3Faculty of Healthy Sciences, Universitat de Vic-Universitat Central de Catalunya (UVic-UCC), 08500 Barcelona, Spain; 4Institute of Translational Pharmacology, National Research Council of Italy (CNR), 90146 Palermo, Italy; giuseppe.insalaco@ift.cnr.it; 5Dentistry, Post-Graduation Program, Evangelical University of Goiás—UniEVANGÉLICA, Anápolis 75083-515, GO, Brazil; orlandoaguedes@gmail.com

**Keywords:** photodynamic therapy, peri-implantitis, photosensitizer

## Abstract

The treatment of peri-implantitis is challenging in the clinical practice of implant dentistry. With limited therapeutic options and drug resistance, there is a need for alternative methods, such as photodynamic therapy (PDT), which is a minimally invasive procedure used to treat peri-implantitis. This study evaluated whether the type of photosensitizer used influences the results of inflammatory control, reduction in peri-implant pocket depth, bleeding during probing, and reduction in bone loss in the dental implant region. We registered the study in the PROSPERO (International Prospective Register of Systematic Review) database. We searched three main databases and gray literature in English without date restrictions. In vivo randomized clinical studies involving individuals with peri-implantitis, smokers, patients with diabetes, and healthy controls were included. PDT was used as the primary intervention. Comparators considered mechanical debridement with a reduction in pocket depth as the primary outcome and clinical attachment level, bleeding on probing, gingival index, plaque index, and microbiological analysis as secondary outcomes. After reviewing the eligibility criteria, we included seven articles out of 266. A great variety of photosensitizers were observed, and it was concluded that the selection of the most appropriate type of photosensitizer must consider the patient’s characteristics and peri-implantitis conditions. The effectiveness of PDT, its effects on the oral microbiome, and the clinical patterns of peri-implantitis may vary depending on the photosensitizer chosen, which is a crucial factor in personalizing peri-implantitis treatment.

## 1. Introduction

Dental implants are among the safest and most reliable alternatives for replacing lost teeth, regardless of the cause, and present high predictability [1,2,3]. This therapeutic modality offers a long-lasting and aesthetically favorable solution that allows patients to recover their chewing function, self-esteem, and confidence when smiling [2]. Like natural teeth, implants are subject to changes in their supporting tissues, including oral pathological conditions, with peri-implantitis being one of the most common [4]. Peri-implantitis is an inflammatory condition that affects the tissues around an implant and can lead to their loss if not treated appropriately [1,4,5,6].

Studies have revealed that the microbiota associated with peri-implantitis mainly comprises gram-negative anaerobic bacteria such as *P. gingivalis*, *A. actinomycetemcomitans*, *T. forsythia*, and *P. intermedia*. Additionally, it may include other bacterial species, such as *Streptococcus* spp. and *Staphylococcus* spp. [7,8,9,10,11,12,13,14,15,16,17,18,19]. These bacteria play a fundamental role in the development and progression of peri-implantitis [10,11].

In addition to microbiological factors, several other factors, such as smoking, diabetes, a compromised immune system, poor oral hygiene, and characteristics related to the implant, can also play a role in the etiology of peri-implantitis [10,20]. The peri-implant disease manifests as symptoms of infection, including suppuration, bleeding, swelling, and redness of the peri-implant tissues, and may present as peri-implant mucositis or peri-implantitis [10,19,21,22,23]. Given the importance of microbial colonization in peri-implantitis, the effective removal of bacterial biofilms from the implant surface is crucial for treatment [24]. Photodynamic therapy (PDT) has emerged as a promising decontamination strategy [25,26,27,28,29,30]. PDT uses the interaction between a light source, a photosensitizer, and oxygen to destroy pathogenic microorganisms selectively, demonstrating effectiveness in reducing the prevalence of pathogens on implant surfaces without harming the implants or surrounding tissues [25,30,31].

However, a challenge PDT faces is the appropriate choice of photosensitizer. A photosensitizer is a molecule that, when activated by light, produces reactive oxygen species in singlet form, which are toxic to microorganisms [20,21,22,26,32]. Methylene blue, indocyanine green (ICG), and toluidine blue are dyes with distinct mechanisms of action used in different therapeutic applications. Methylene blue, composed of the phenothiazinium cation, reacts with oxygen in activated cells, generating Reactive Oxygen Species (ROS) that cause oxidative damage, including lipid oxidation, protein damage, and DNA modifications. In turn, ICG, when activated by light at a specific frequency, generates heat and ROS, being selectively absorbed by target cells and subsequently activated by light for localized treatments. Meanwhile, toluidine blue, when activated by light at a specific frequency, transfers energy to molecular oxygen in cells, forming ROS that cause oxidative and direct damage to target cells, contributing to the efficacy of PDT [21,22,26,32].

The selection of an ideal photosensitizer is crucial, as it must be able to selectively bind to pathogenic microorganisms without affecting healthy host cells. Furthermore, the photosensitizer must be stable, non-toxic, and have a good light absorption capacity in the appropriate wavelength range. Therefore, research and development of new photosensitizers with these characteristics are necessary to improve the effectiveness of PDT in implant decontamination. This systematic review aimed to investigate the influence of the photosensitizer type on the treatment of peri-implantitis using PDT. The results of this study are expected to provide an in-depth understanding of the effectiveness of this approach and contribute to the improvement of therapeutic protocols for peri-implantitis.

## 2. Materials and Methods

This systematic review was conducted following the Preferred Items for Reporting of Systematic Reviews and Meta-Analyses (PRISMA) guidelines [33,34,35] (Appendix A) and registered at the International Prospective Register of Systematic Reviews (PROSPERO), under the number CR42023473608.

### 2.1. Eligibility Criteria

Randomized clinical trials that included participants from different groups, such as patients with peri-implant diseases, smokers, patients with diabetes, and healthy controls. The main intervention evaluated was PDT, which could be used as both main and adjuvant therapy. The comparators or control groups underwent mechanical debridement, either surgical or nonsurgical. The primary outcome analyzed was the reduction in pocket depth (PD); in contrast, the secondary outcomes included clinical attachment level, bleeding on probing (BOP), gingival index (GI), plaque index, and microbiological analysis. In-vitro studies, literature reviews, letters, opinions, case reports, case series, and abstracts were excluded from the analysis.

### 2.2. Search Information

An electronic literature search was conducted in PubMed/Medline, Web of Science, and Scopus without restrictions regarding the year of publication in August 2023. In addition, a manual search of the references of studies and experts was conducted to locate any publications not identified electronically.

### 2.3. Search Algorithms

The following keywords and their combinations were used: “Photochemotherapy”[Mesh] OR “Photochemotherapies” OR “Photodynamic Therapy” OR “Photodynamic Therapies”; AND “Peri-Implantitis”[Mesh] OR “Peri Implantitis” OR “Peri-Implantitides” OR “Periimplantitis” OR “Periimplantitide”; AND “Photosensitizing Agents”[Mesh] OR “Photosensitizers” OR “Photosensitizer” OR “Photosensitizing Agent” OR “Photosensitizing Effect” OR “Photosensitizing Effects”.

### 2.4. Studys Selection

A two-phase process was adopted to select studies. In phase 1, two reviewers (TBMOS and JPRA) independently screened the titles and abstracts to identify eligible studies based on the eligibility criteria. Those who met the inclusion criteria were selected for full reading. In phase 2, the same reviewers independently read the studies in full to confirm inclusion. Any disagreement between the two reviewers was resolved through discussion with a third reviewer (LVFO) when necessary.

### 2.5. Data Extraction

The same double-review process was performed independently to collect all data and for subsequent comparisons. Disagreements at this stage were resolved through discussion, and if necessary, a final consensus was reached with the help of a third reviewer. The descriptive characteristics of all included studies were extracted, such as study characteristics (authors and year), country of completion, study design, inclusion and exclusion criteria, clinical parameters and collection points, photosensitizer use, intervention, and results (Table 1, Table 2 and Table 3). If the necessary data were incomplete or missing, the authors were contacted to access the missing information.

### 2.6. Risk of Bias in Individual Studies

The updated Consolidated Standards of Reporting Trials statement guidelines were adopted to assess the methodological quality of the included randomized controlled trials (RCTs) [43]. To verify the validity of the eligible RCTs, we examined the risk of bias related to allocation concealment, randomization, masking of the outcome assessor, and masking of patients. The Cochrane Manual for Systematic Reviews of Interventions [44] was used to assess the possibility of bias in each study, classifying it as “high risk of bias” (high), “low risk of bias” (low) or “uncertain” (?) in each section. In general, studies were categorized as (i) low risk of bias if all criteria were met (adequate randomization and allocation concealment; answer “yes” to all questions about the integrity of outcome data and blinding, and “no” response to selective reporting and other sources of bias); (ii) uncertain risk of bias if one or more criteria were partially met; or (iii) high risk of bias if one or more criteria were not met.

## 3. Results

Based on the titles and abstracts, 359 studies were initially identified. After removing duplicates (n = 93) and screening the abstracts, 244 articles that did not meet the review inclusion criteria were excluded. Twenty-two full-text studies were selected for the evaluation, of which fifteen were excluded because they did not meet the inclusion criteria (Table 4). The final selection resulted in the inclusion of seven studies [36,37,38,39,40,41,42], all of which used PDT for the treatment of peri-implantitis (Table 1, Table 2 and Table 3). Figure 1 illustrates the flow of the study selection process, and the literature search results according to the PRISMA guidelines [33].

### 3.1. Study Characteristics and Results of Individual Studies

Of the seven included studies, five were randomized clinical trials [37,39,40,41,42], one was a randomized controlled trial [38], and one was a randomized double-blinded clinical trial [36]. These studies were conducted in different locations, including Saudi Arabia [37,38,39,40], Iran [36,42], and Japan [41], and involved a variable number of participants, with samples ranging from 10 to 64 patients and ages ranging from 20 to 90 years. Of the 230 participants, 163 were males and 67 were females. However, one of the articles [41] did not mention how many patients participated or their sex; it only reported that 15 implants were analyzed in the study. In two articles [37,38], all participants were males.

Some patients had a history of smoking in two studies [37,38], and three studies [37,39,40] included patients diagnosed with diabetes mellitus, one of which examined the relationship between type 2 diabetes and smoking. Three studies [36,41,42] analyzed peri-implantitis without considering risk factors. The criteria for diagnosing problems around implants varied among studies. Six of them [36,37,38,39,40,42] measured outcomes such as peri-implant PD, BOP, plaque accumulation (PI), bone loss at buccal sites or lingual/palatal implants (CBL), gingival health (GI), clinical attachment loss (CAL), gingival recession (MR), and the presence of pus.

Of the seven studies, only one [41] did not perform manual or nonsurgical debridement before treatment. Four studies incorporated oral hygiene instructions [36,38,39,42], and one [39] used mouthwashes with 0.12% chlorhexidine gluconate. All studies followed patients for a period ranging from 7 days to 6 months.

### 3.2. Risk of Bias

Of the seven studies, only one [30] was considered low quality; in contrast, two [37,38] were considered unclear. The results of the risk-of-bias assessment are presented in Figure 2.

### 3.3. Certainty of Evidence

According to the GRADE criteria [44], reliance on cumulative evidence for pairing PD outcomes was classified as high quality (Figure 3). This is due to the exclusive use of RCTs in the systematic review, which substantially reinforces the reliability of the conclusions. However, the evaluation of the effectiveness of PDT resulted in a moderate evidence rating because of the inconsistencies in the results presented in the analysis of its effectiveness.

### 3.4. Photosensitizers

Of the seven studies, diverse photosensitizers were observed. Two studies used methylene blue (MB) [38,39], two used indocyanine green (ICG) [31,41], two used toluidine blue (TB) [41,42], one used phenothiazine chloride [37], one used the photosensitizer EmunDO [36], and one used methylthionine chloride (MTC) [41]. In addition, one study used two photosensitizers to evaluate the different effects [39], comparing MB and ICG.

## 4. Discussion

### 4.1. Sounding Depth Index

PD was the main criterion for evaluating the effectiveness of intervention with photosensitizer-mediated PDT (GFR). Among the six articles that measured PD [36,37,38,39,40,42], two did not report statistically significant improvements [36,37] when comparing baseline values with follow-up measurements or between-group comparisons between testing and controls. However, one of these studies [36] mentioned improvements in all clinical parameters. In contrast, the other four studies [38,39,40,42] showed a statistically significant improvement in PD. In one of the studies, Al Rifaiy et al. [38] reported a reduction in PD (*p* < 0.001) in the test group compared to the control group after 12 weeks. In another study [39], a reduction in PD was observed, with values of 0.45 ± 0.41 mm in the control group, compared to 0.84 ± 0.62 mm in the test group with ICG-mediated GFR and 0.85 ± 0.63 mm in the test group with MB-mediated GFR. Furthermore, two additional studies [40,42] also indicated a significant reduction in PD in all groups tested at each follow-up visit when compared with baseline values (*p* < 0.05). Therefore, these results suggest a general trend of improvement in PD due to photosensitizer-mediated PDT intervention; however, some studies have found variations in results.

### 4.2. Gingival Bleeding Index

BOP is a secondary clinical parameter widely used to evaluate the clinical evolution of PDT intervention in cases of peri-implantitis. However, among these studies [36,37,38,39,40,41,42] analyzed, variations in results related to BOP were observed. Birang et al. [36] highlighted improvements in the clinical conditions around implants with peri-implantitis during short follow-up periods with no significant difference between the test and control groups. In another study [39], which compared the effects of ICG and MB, the bleeding rate in the control group was 12.10 ± 19.30%; in contrast, in the group undergoing intervention with PDT mediated by ICG, it was 28.77 ± 29.24%, and in the group with MB-mediated PDT it was 27.71 ± 28.16%. These results demonstrate improvements in the bleeding pattern upon probing for both photosensitizers, indicating comparable results. In contrast, Karimi et al. [42] reported achieving complete resolution of BOP in 100% of implants in the test group within 3 months. Furthermore, the study by Elsadek [40] pointed to a statistically significant reduction in BOP in all groups tested during each follow-up visit compared with baseline values (*p* < 0.05).

However, other studies [37,38] reported no statistically significant differences in BOP between groups compared to baseline values [37,38]. In the study by Al Rifaiy et al. [38], the values found for the test group were 53.3 ± 4.2% (initial) and 48.2 ± 3.6% (6 months of follow-up); in contrast, in the control group, they were 35.2 ± 3.1% (initial) and 33.1 ± 2.4% (6 months of follow-up). These variations in BOP results highlight the complexity and possible heterogeneity of the response to PDT in patients with peri-implantitis.

### 4.3. Plaque Index

The plaque index is an evaluation parameter presented in five studies analyzed [36,38,39,40,41]. In one study [39], significant changes were observed in the plaque index averages between the beginning of the study and the 3-month follow-up, with these differences being statistically significant between the control group (12.42 ± 21.80%) and the test groups that received PDT mediated by ICG (26.55 ± 25.80%) and PFT mediated by MB (27.24 ± 26.15%). This indicated an improvement in the oral health of these groups after the intervention. In contrast, two studies [36,38] reported no statistically significant differences in the plaque index when comparing follow-up data with baseline values despite observing improvements in clinical bleeding patterns after the intervention. In other studies [40,41], an improvement in inflammatory patterns was reported after the intervention, suggesting a positive impact of PDT on the control of bacterial plaque and oral health in general.

### 4.4. Microbiological Analyzes

In a study conducted in Iran by Birang et al. [36], laser PDT significantly reduced the *P. gingivalis* count. This is an important finding because *P. gingivalis* is associated with periodontal and peri-implant diseases [36]. However, the difference in bacterial reduction between the treatment groups was not statistically significant, suggesting that PDT, although promising, did not significantly outperform conventional treatment. In contrast, a study conducted in Saudi Arabia [39] focused on patients with type 2 diabetes and peri-implantitis. In this study, PDT with indocyanine green and methylene blue led to notable microbiological improvements. Specifically, there was a significant reduction in the number and proportion of *T. forsythia*, a strain of harmful periodontal bacteria. However, the study did not show statistically significant differences in the other bacteria evaluated, such as *F. nucleatum*, *P. intermedia*, and *T. denticola*, between the treatment groups.

Although both studies demonstrated a positive trend toward reducing the bacterial load associated with peri-implantitis through PDT, it is worth noting that the different treatment protocols and sample characteristics may explain some of the differences in the microbiological results observed. Furthermore, the microbial analysis suggested that PDT may have differential effects on specific bacteria, such as *P. gingivalis* and *T. forsythia*, highlighting the need to consider the mechanisms of action of PDT on different pathogens. Both studies used PDT as an adjuvant in treating peri-implantitis, with notable microbiological improvements. However, bacterial efficacy must be contextualized for specific patient characteristics, treatment protocols, and target bacteria. These findings highlight the need for further studies to more fully understand PDT’s role in managing peri-implantitis, its effectiveness in different clinical contexts, and how it can be optimized to improve microbiological results.

### 4.5. Comparative Results

The results of four studies indicated a statistically significant improvement in clinical standards [36,39,40,41]. Furthermore, in two studies [36,38], PDT adjuvant to manual debridement proved more effective than manual debridement alone. In contrast, only one study [37] reported finding no statistically significant differences in the use of PDT for treating peri-implantitis. This suggests that PDT can be an effective approach in the search for improved results; however, there are variations in results between studies.

### 4.6. Efficacy of PDT in Smoking Patients

Abduljabbar et al., in 2017 [37], carried out a study in Saudi Arabia based on patients with type 2 diabetes who smoke and the treatment of peri-implantitis with PDT. The results showed no statistically significant differences in PD or bleeding rate between smokers with type 2 diabetes and non-smokers under the same conditions after PDT treatment. These findings suggest that, in the short term, PDT may not offer a substantial additional benefit in treating peri-implantitis in smokers compared with conventional mechanical debridement. In contrast, a study conducted by Al Rifaiy et al. [38] in Saudi Arabia was based on patients who smoked electronic cigarettes (vaporizers). The results of this study revealed that PDT adjuvant to mechanical debridement was effective in improving peri-implant clinical parameters compared to mechanical debridement alone. These findings may suggest that PDT may be more beneficial in patients who smoke electronic cigarettes compared to those who use traditional cigarettes. In addition to these studies, the work of Karimi et al. [42], carried out in Iran, evaluated the effectiveness of PDT in the treatment of peri-implant mucositis and peri-implantitis in patients who smoke. The results of this study demonstrated significant improvements in peri-implant clinical parameters, including probing PD (PPD), clinical attachment loss (CAL), BOP, and GI in patients treated with PDT compared with mechanical debridement. This suggests that PDT may be an effective option for the treatment of peri-implant diseases in smokers.

In summary, the effectiveness of PDT in the treatment of peri-implantitis in patients who smoke may vary according to the type of smoking, whether conventional or electronic. PDT appears to be more beneficial for patients who use e-cigarettes [38] compared to those who smoke traditional cigarettes [37]. Karimi et al. [42] highlighted the potential of PDT for the treatment of peri-implant diseases in smokers, regardless of the type of smoking. Therefore, it is crucial to consider smoking status when evaluating the effectiveness of PDT and tailoring treatment protocols according to individual patient needs.

### 4.7. Efficacy of PDT in Diabetic Patients

Abduljabbar et al. [37] investigated the effectiveness of PDT in patients with type 2 diabetes. The results of this study showed no statistically significant differences in peri-implant clinical parameters between patients with type 2 diabetes and those without type 2 diabetes after treatment with PDT and mechanical debridement. This suggests that, in the short term, PDT may be a viable option for treating peri-implantitis in patients with diabetes, with results comparable to those in patients without diabetes. However, it is essential to highlight the study conducted by Elsadek et al. [40], which analyzed the efficacy of adjuvant PDT compared to treatment alone in diabetes mellitus (DM2) in patients with peri-implantitis and DM2. The results of this study indicated comparable results in terms of peri-implant clinical parameters and proinflammatory characteristics between the use of adjuvant PDT and treatment with DM alone in diabetic patients. This finding suggests that the effectiveness of PDT in patients with diabetes varies according to treatment protocol and patient selection.

Furthermore, Alsayed et al. [39] evaluated patients with type 2 diabetes and peri-implantitis by comparing PDT performed with indocyanine green (ICG) and methylene blue for treating DM alone. The results indicated complete resolution of BOP within 3 months, which was achieved in 100% of implants in the group treated with ICG-mediated PDT. Furthermore, there were improvements in the mean probing depth measurements and CAL gains in implants in the group treated with PDT compared to those treated with DM alone. These findings suggest that PDT, especially when mediated by indocyanine green, may effectively treat peri-implantitis in patients with type 2 diabetes. In contrast, a study conducted in Iran in 2016 by Karimi et al. [42] focused on the effectiveness of TFP in treating peri-implant mucositis and peri-implantitis in patients without uncontrolled diabetes. The results demonstrated significant improvements in peri-implant clinical parameters, including probing PD (PPD), CAL, BOP, and GI, in patients treated with PDT compared with mechanical debridement. This evidence suggests that PDT may be an effective treatment option for peri-implant diseases in patients without diabetes.

In summary, the studies examined present a comprehensive overview of PDT’s effectiveness in treating peri-implantitis in patients with diabetes. The variability in the results highlights the importance of considering treatment protocols, patient selection, and underlying conditions when evaluating the effectiveness of PDT in patients with diabetes. Additional research is needed to understand the role of PDT better and establish more precise clinical parameters for treating peri-implantitis in patients with diabetes.

### 4.8. Side Effects and Safety

Based on the selected studies, it can be concluded that careful selection of participants is essential to guarantee the safety of PDT. Most studies established inclusion criteria that excluded patients with medical conditions that could increase risks, such as uncontrolled diabetes, recent antibiotic use, pregnancy, breastfeeding, and other conditions [37,39,40,41,42]. Furthermore, these studies used different photosensitizers and irradiation parameters, which may have influenced the safety and efficacy of the therapies. The choice of photosensitizer and irradiation parameters must be carefully evaluated to determine the most appropriate regimen for each patient [36,37,38,39,40,41,42]. Studies have not reported severe adverse events related to PDT; however, there have been reports of temporary discomfort during or after the procedure, such as sensitivity to light and heat. These effects are generally transient and short-lived [37,38,39,41,42]. Most studies had relatively short follow-up periods, and the long-term safety of PDT requires further evaluation. It is essential to conduct long-term follow-up studies to determine the stability of the results and identify late side effects [37,39,40,41,42]. Despite this, most studies have demonstrated significant improvements in clinical, microbiological, and radiographic parameters in patients undergoing PDT compared to conventional treatment, suggesting that the clinical benefits obtained with PDT may outweigh any risks or mild side effects associated with the therapy [36,38,39,40,41,42].

In summary, PDT appears to be a promising option for treating peri-implantitis, with mild and temporary side effects, as long as qualified professionals perform it in carefully selected patients. However, further research is required to ensure its long-term safety.

### 4.9. Study Limitations

The limitations of the reviewed studies are evident and must be considered when extrapolating the results obtained. Many studies had small sample sizes, which may have limited the generalizability of their findings to a broader population. Furthermore, the methodological quality of the studies varied, with some not reaching a low risk of bias based on the tools used. These limitations may have reduced the robustness of our conclusions. An important point worth highlighting is the need for bacterial quantification in the methodologies used in previous studies. Most studies have focused on clinical and microbiological assessments and have not comprehensively analyzed bacterial load reduction. Bacterial quantification is essential for proving the effectiveness of PDT in terms of bacterial reduction. Therefore, the lack of data in this regard can be considered a critical limitation as it prevents a comprehensive assessment of the impact of PDT on peri-implant infections.

Furthermore, the variation among studies regarding treatment protocols, choice of photosensitizer, and irradiation parameters also represents a limitation. This variation makes a direct comparison of the results difficult and highlights the need to establish clearer clinical guidelines for using PDT in treating peri-implantitis. Although the reviewed studies offer promising evidence for using PDT in treating peri-implantitis, it is essential to recognize the limitations of the available research. To improve the understanding and clinical application of PDT, it is necessary to conduct new studies with larger samples, greater methodological rigor, and a focus on bacterial quantification. These measures will help establish a stronger foundation for the effectiveness of PDT in managing peri-implantitis and guide clinical practice more precisely.

## 5. Conclusions

When investigating the influence of the photosensitizer on the success of peri-implantitis treatment with photodynamic therapy (PDT) through a systematic review, we concluded that the choice of photosensitizer plays a crucial role in treating peri-implantitis using PDT. Several photosensitizers, such as methylene blue and indocyanine green, have been used in these studies. This emphasizes the importance of selecting the most appropriate photosensitizer based on selectivity, patient characteristics and peri-implantitis conditions.

However, it is necessary to standardize intervention protocols and conduct long-term follow-up studies to determine the durability of the findings and evaluate the real effectiveness of PDT compared to conventional treatment with manual debridement. Additionally, it is essential to further discuss specific patient factors, such as genetics, immune status, and oral hygiene, in order to better predict how these factors influence PDT outcomes. This information is crucial for personalized treatment planning and predicting the success of peri-implantitis treatment with PDT.

## Figures and Tables

**Figure 1 pharmaceutics-16-00307-f001:**
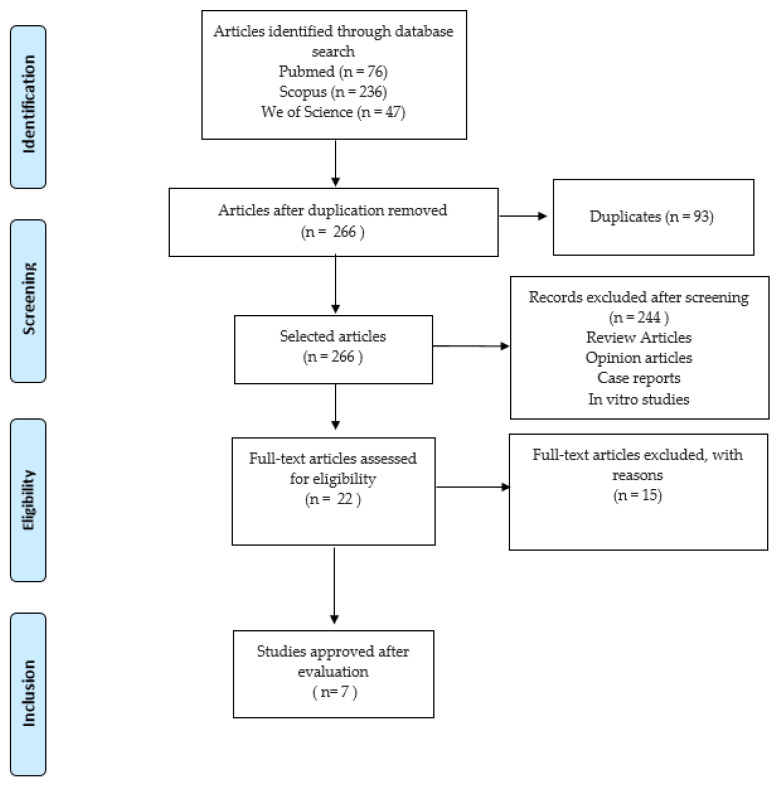
Flow diagram of the current systematic review conducted according to the Preferred Reporting Items for Systematic Reviews and Meta-analysis (PRISMA) guidelines.

**Figure 2 pharmaceutics-16-00307-f002:**
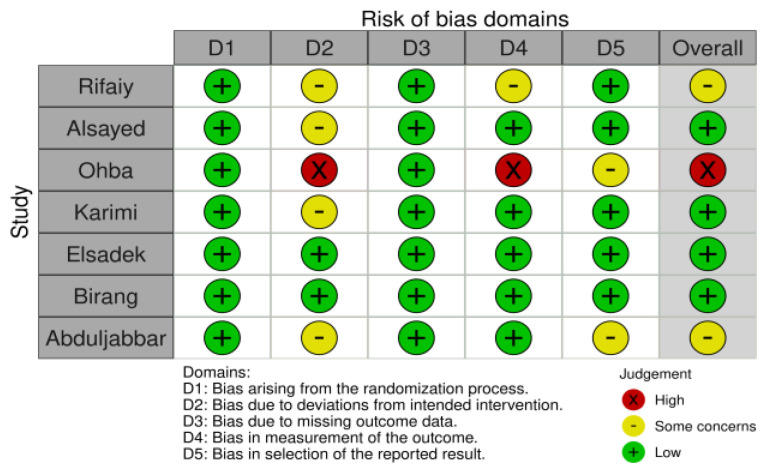
Risk of bias summary: review authors’ judgments about each risk of bias item for each included study.

**Figure 3 pharmaceutics-16-00307-f003:**
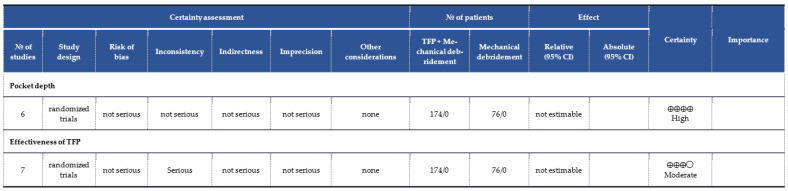
Risk of bias summary: review authors’ judgments about each risk of bias item for each included study. Question: TFP + Mechanical debridement compared to mechanical debridement for patients with peri-implantitis.

**Table 1 pharmaceutics-16-00307-t001:** Information from studies utilizing EmunDo and phenothiazine chloride as photosensitizers.

Author/Study Design	Clinical Parameters	Photosensitizer/Intervention	Results
Birang et al., (2017) [36] **STUDY DESIGN** n = 20, 10 M/10 F20 to 67 years old MI = 36.6 ± 9.7 years Group 1—PDT Group 2—DM + Lasertherapy Double-blind randomized clinical study Lasertherapy × PDT in individuals with primary peri-implantitis Follow-up: 6 weeks and 3 months	Probing pocket depth (PPD), papilla bleeding index (BOP), modified plaque index (PI) **COLLECTION POINTS** Collection points: distobuccal (DV), mesiobuccal (MV), distolingual (DL), mesiolingual (ML) Collection: initial, 6 weeks and 3 months	EmunDo **INTERVENTION** Initial—manual debridement (DM) with ultrasound (US) and DM with plastic curette, sodium bicarbonate jet in G1 and G2 **G1**—US + DM + PDT with EmunDo 90 s + irrigation with 0.9% saline solution + transgingival irradiation 30 s/300 mW + intra-pocket irradiation 30 s/300 mW + elimination of intra-pocket granulation tissue 30 s/300 mW 15 days—repeat the initial intervention Instructions for participants: Oral hygiene guidance (OHG)	G1 and G2 showed statistically significant improvements in bleeding on probing (*p* < 0.001), probing pocket depth (PPD) (*p* = 0.006) and modified plaque index (*p* < 0.001), without significant differences between the groups. Two groups (*p* > 0.05) and *P. gingivalis* (*p* = 0.015) in the control group significantly decreased. Laser therapy only significantly decreased *P. gingivalis* (*p* = 0.015) and differences in *A. actinomycetemcomitans* were threshold significant (*p* = 0.061). PDT significantly decreased *A. actinomycetemcomitans* (*p* = 0.022), *T. forsythia* (*p* = 0.038) and *P. gingivalis* (*p*= 0.050). Mann-Whitney test revealed no significant difference in changes in bacterial counts before and after treatment between treatment modalities (*A. actinomycetemcomitans p* = 0.846, *P. gingival p* = 0.503, *P. intermediate p* = 0.682, *T. denticola p* = 0.399 and *T. forsythia p* = 0.199)
Abduljabbar (2017) [37] **STUDY DESIGN** n = 64, all men G1—n = 33 DM2 smokers/MI = 52.6 ±0.8 years G2—n = 31 DM non-smokers/MI = 54.4 ± 1.2 years Randomized clinical study with 6-month follow-up	BO, PD ≥ 4 mm **COLLECTION POINTS** DV, MV, DL, ML, V, P/L	Phenothiazine chloride **INTERVENTION** DM with US + adjuvant PDT with phenothiazine chloride 120 s + pocket irrigation with 3% hydrogen peroxide + irradiation with 600 nm/100 mW diode laser for 10 s in a single session + high-power liquid chromatography (HbA1c) at baseline and after 6 months (G1 + G2) Instructions to participants not reported	At baseline, BOP and PD ≥ 4 mm were comparable between subjects in groups 1 and 2. At 6-month follow-up, there was no statistically significant difference in BOP and PD ≥ 4 mm between patients in groups 1 and 2 compared to respective baseline values. HbA1c levels were comparable in all groups at all time intervals. HbA1c at baseline and 6 months later—G1—9.3% (start)/8.4 (6 months)—G2—8.7% (start)/8.4 (6 months)—BOP—G1—53, 3 ± 4.2% (start)/48.2 ± 3.6% (6 months)—G2—35.2 ± 3.1% (start)/33.1 ± 2.4% (6 months)—PD—G1—26.2 ± 3.7% (start)/25.1 ± 0.8% (6 months)—G2—29.5 ± 3.7% (start)/25.5 ± 1.4% (6 months)

**Table 2 pharmaceutics-16-00307-t002:** Information from studies utilizing methylene blue, indocyanine green and methylthionine chloride as photosensitizers.

Author/Study Design	Clinical Parameters	Photosensitizer/Intervention	Results
Al Rifaiy et al., (2018) [38] **STUDY DESIGN** n = 38, all men G1—n = 20 MI = 33.6 ± 2.8 years/DM + PDT G2—n = 18 MI = 35.4 ± 2.1 years/DM Randomized controlled clinical trialFollow-up for 12 weeks	IP, BOP, PD **COLLECTION POINTS** DV, MV, DL, ML, buccal(V), palatal/lingual (P/L)	Methylene blue (TB) 0.005% **INTERVENTION** **G1**—DM + PDT with TB applied to the periodontal pocket with a 10 s wait + 60 s irradiation with a 670 nm/150 mW diode laser in a single application **G2**—DM with plastic curette Instructions for participants: OHG, guidelines for quitting smoking	Start—PI, BOP and peri-implant PD comparable between G1 and 2. After 12 weeks, statistically significant reduction in PI (*p* < 0.001) and PD (*p* < 0.001) between G1 and G2 patients compared to baseline. Significant reduction in IP (*p* < 0.001) and PD (*p* < 0.001) for G1 compared to G2 at 12 weeks. There was no statistically significant difference for BOP between groups at follow-up
Alsayed et al., (2023) [39] **STUDY DESIGN** n = 60, 35 M/25 F G1—n = 20/MI = 56.5 ± 6.6 years/DM G2—n = 20/MI = 53.4 ± 4.8 years/DM + PDT with ICG G3—n = 20/MI = 57.5 ± 4.1 years/DM + PDT with MB Randomized clinical trial Follow-up for 3 months	IP, PD, BOP, crest bone level (CBL) **COLLECTION POINTS** DV, MV, DL, ML, V, P/L CBL collection by bitewing digital radiographs	Indocyanine green (ICG) and methylene blue (MB) **INTERVENTION** **G1**—DM with plastic curette **G2**—DM + PDT with ICG 1 mg/mL, 60 s wait + irrigation (saline solution) + irradiation: (a) photobiomodulation tip in the 30 s papilla (6J), (b) tip of the bulb in the 10 s palatine groove and sulcus bottom lingual to coronal (4J), 810 nm/200 mW (continuous mode), single application **G3**—DM + PDT with MB, 60 s wait + irrigation (saline solution) + irradiation with 670 nm/140 mW (21 Jcm^2^), single application Instructions for participants: OHG, guidelines for using mouthwash with chlorhexidine gluconate (0.12%), twice a day, 60 s, to avoid the consumption of anti-inflammatories during the study	Mean changes between baseline and 3-month follow-up in peri-implant clinical-radiographic parameters were significantly different between control (PI: 12.42 ± 21.80%; BOP: 12.10 ± 19.30%; PD: 0.45 ± 0.41 mm; CBL:1.10 ± 1.02 mm) and test groups (ICG-mediated PDT [PI: 26.55 ± 25.80%; BOP: 28.77 ± 29.24%; PD: 0.84 ± 0.62 mm; CBL: 1.98 ± 1.85 mm] and MB-mediated PDT [PI: 27.24 ± 26.15%; BOP: 27.71 ± 28.16%; PD: 0.85 ± 0.63 mm; CBL: 1.95 ± 1.80 mm]); comparable differences observed in peri-implant PI, BOP, PD and CBL between G2 and G3 participants (*p* > 0.05). The proportions of *T. forsythia* were significantly reduced in G2 (4.78 × 104 CFU/mL) and G3 (4.76 × 104 CFU/mL) as compared to G1 (−4.40 × 103 CFU/mL) at 3-month follow-up (*p* = 0.02). No statistically significant differences were observed between the study groups in relation to the proportions of the other target bacteria species evaluated. For IL-6 (G 1: 210 ± 108; G2: 298 ± 165; G3: 277 ± 121 pg/mL; *p* = 0.03), IL-1β (G1: 101 ± 95; G2: 84 ± 98; G3: 86 ± 74 pg/mL; *p* = 0.02) and TNF-α (G1: 36 ± 121; G2: 385 ± 210; G3: 366 ± 198 pg/mL; *p* = 0.03) at PISF levels, a statistically significant reduction at the 3-month follow-up.
Elsadek (2023) [40] **STUDY DESIGN** n = 38 G1—n = 13/5 men and 8 women/DM + PDT with ICG/MI = 45.3 ± 3.9 years; G2—n = 12/3 men and 9 women/DM + PDT with TCM/MI = 47.6 ± 6.5 years; G3—n = 13/6 men and 7 women/DM/MI = 48.2 ± 7.8 years Randomized clinical trial Follow-up: 3 months and 6 months	Plaque scores (PS), peri-implant probing scores (PPS), bleeding scores (BS), estimated peri-implant bone loss (PIBL) linearly from 2 mm below the abutment interface to the most level of the crest of the bone. **COLLECTION POINTS** Three lingual and three buccal surfaces.	**G1**—Indocyanine green solution (ICG) 1 mg/mL/**G2**—methylthionine chloride (MTC) **INTERVENTION** **G1**—DM with US and scaling and root planing with curette + PDT with ICG with a wait of 60 s + irradiation 810/300 mW, with a fluence of 56 Jcm^2^ pulsed mode 30 s with continuous vertical movement, single application **G2**—DM with US and scaling and root planing with curette + PDT with MTC waiting for 60 s + removal of excess dye gently + irradiation with a 660 ± 10 nm/100 mW diode laser with the tip inserted into the depth of the pocket and moved circumferentially around the implant, 120 s/location. Each irradiation point about 0.4 cm^2^, radiant exposure 30 Jcm^2^, irradiance 0.25 Wcm^2^ 1 point every 3 mm, single application Instructions to participants not reported	Significant reduction for PS, BS and PPS in all groups tested at each follow-up visit compared to baseline values (*p* < 0.05). Substantial decrease in PIBL in all patients in the group at the 6-month follow-up compared to the 3-month follow-up (*p* < 0.05). Levels of IL-6 and TNF-α, substantial reduction in all study groups up to 6 months from their initial scores (*p* < 0.05). No changes in AGEs levels were observed in any group at any of the visits (*p* > 0.05).

**Table 3 pharmaceutics-16-00307-t003:** Information from studies utilizing toluidine blue as photosensitizers.

Author/Study Design	Clinical Parameters	Photosensitizer/Intervention	Results
Ohba et al., (2020) [41] **STUDY DESIGN** n = 21 implants G1—n = 13 implants/irrigation with saline solution G2—n = 12 implants/PDT 20–90 years Randomized clinical trial Follow-up for 7 ± 2 days	Pus discharge volume, BOP, IP **COLLECTION POINTS** Not mentioned	Toluidine blue (TB) 0.1 mg/mL **INTERVENTION** TB 0.1 mg/mL with 630 nm LED (620–640 nm), in two applications in the same session **G1**—irrigation with 5 mL saline solution **G2**—irrigation with 5 mL saline solution + PDT with TB 0.1 mg/mL without waiting time + irradiation with 630 nm LED (620–640 nm buccal/labial and lingual/palatal sides 30 s + irrigation with 5 mL of solution saline. Repeat the procedure once more, in the same session. Evaluations in 7 ± 2 days Instructions to participants not reported	Pus discharge decreased in 7 of 12 implants (58.3%) in G2 and in 2 of 13 implants (15.4%) in G1. Fisher’s exact test—PDT resulted in a statistically significant decrease in pus discharge compared to irrigation alone (*p* = 0.0414)”
Karimi et al., (2016) [42] **STUDY DESIGN** n = 30 implants, 10 individuals 2 M and 8 F n = 15 implants/group G1—DM G2—DM + PDT/MI = 52.8 years Randomized clinical trial Follow-up for 3 months	Gingival index (GI), BP, PD, MR, clinical attachment loss (CAL) **COLLECTION POINTS** PPD and CAL DV, MV, DL, ML, V, P/L, buccal and lingual	Toluidine blue (TB) 0.01% **INTERVENTION** TB 0.01% 180 s with 630 diode laser nm/2.00 mW/cm^2^ in single application **G1**—DM with plastic curette + irrigation of the pocket with sterile saline solution **G2**—DM with plastic curette + irrigation of the pocket with sterile saline solution + PDT with TB waiting time 180 s + irradiation in 6 niches, for 20 s each (total 120 s). Instructions for participants: Individualized OHG, depending on the type of prosthesis Clinical parameters measured immediately before treatment and re-evaluated 1.5 and 3 months after treatment, using a plastic probe	Significant differences in PPD, CAL, BOP, and GI at each time point between the two groups. There were no statistically significant changes in relation to any of the control group parameters. Complete resolution of BOP within 3 months achieved in 100% of test implants. At 1.5 and 3 months, there were differences in the mean probing depth and CAL gain measurements in the implants in the test group

**Table 4 pharmaceutics-16-00307-t004:** Deleted articles and reason for exclusion—n = 15.

Author, Year	Reason for Exclusion
1. Aabed et al., 2022 [45]	1
2. Abdellatif et al., 2022 [46]	1
3. Abduljabbar, 2017 [47]	2
4. Afrasiabi et al., 2022 [48]	3
5. Ahmed et al., 2022 [49]	4
6. Ahmed et al., 2020 [4]	5
7. Al Amri et al., 2016 [50]	6
8. Al Deeb et al., 2020 [51]	6
9. Al Deeb et al., 2020 [52]	7
10. Al-Khureif et al., 2020 [53]	5
11. Herbert et al., 2013 [54]	10
12. Dörtbudak et al., 2001 [55]	8
13. Albaker et al., 2018 [56]	9
14. Harmouche et al., 2019 [57]	1
15. Javed et al., 2017 [58]	6

1—Individuals with periodontitis; 2—individuals with pre-diabetes; 3—individuals with oral infection; 4—evaluation of cytokine levels; 5—adjuvant antibiotic therapy; 6—individuals with peri-implant inflammation; 7—assessment of bone biomarker levels; 8—evaluation of microbial samples, 9—associated surgical debridement; 10—assessment of bone defects.

## Data Availability

The data presented in this study are available in this article (and Appendix A).

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
