# Peer review of "Influence of Photosensitizer on Photodynamic Therapy for Peri-Implantitis: A Systematic Review"

_pharmaceutics, 2024, doi:10.3390/pharmaceutics16030307_

Round 1

Reviewer 1 Report

Comments and Suggestions for Authors

Dear authors,

I have few comments and suggestions:

1. there are too many authors for this study...

2. parts of the text in the methodology are too extensive, please shorten them

3. please reduce the content or the number of the tables in your manuscript

4. there are photosensitizers that you have not described, such as riboflavin...

5. the number of references is too high, there are too many references related to periodontitis

Author Response

Reviewer's comments

Answers

We appreciate all suggestions. Modifications made to the text were highlighted in red.

1. there are too many authors for this study...

Ok. The number of authors has been reduced. The current authors collaborated effectively in the design, execution, review and guidance of the study.

2. parts of the text in the methodology are too extensive, please shorten them.

Ok. The text has been modified in accordance with the suggestions provided by the reviewer.

3. please reduce the content or the number of the tables in your manuscript.

Ok. The number of tables and the information presented therein have been modified in accordance with the recommendations provided by the reviewer.

4. there are photosensitizers that you have not described, such as riboflavin...

Riboflavin holds fundamental importance in aerobic organisms as a precursor of crucial coenzymes involved in the electron transport chain. It has garnered attention in the field of oncology. However, our literature review for this study did not yield any reports on the use of riboflavin, possibly due to the predominantly anaerobic composition of the microbiota present in peri-implantitis.

5. the number of references is too high, there are too many references related to periodontitis.

The inclusion of a high number of references is necessary, especially for a systematic review. Given that periodontitis is a highly prevalent pathology closely linked to peri-implantitis, articles related to this topic were incorporated into the text.

Reviewer 2 Report

Comments and Suggestions for Authors

In this manuscript, “Influence of Photosensitizier on Photodynamic Therapy for Peri-Implantitis: A Systematic Review” by Schweigert et al. evaluates whether the type of photosensitizer used influences the results of inflammatory control, reduction in peri-implant pocket depth, bleeding during probing, and reduction in bone loss in the dental implant region. Although, some literature is collected, some comparison in Table is not easy to understand. Therefore, I would suggest authors may revise or add some more Tables or Schemes before resubmission. Here are the comments and suggestions:

1.     Please revise Table 1, it is not easy to read and compare each other.

2.     The conclusions should be extended and please add perspective.

Author Response

Reviewer's comments

Answers

In this manuscript, “Influence of Photosensitizier on Photodynamic Therapy for Peri-Implantitis: A Systematic Review” by Schweigert et al. evaluates whether the type of photosensitizer used influences the results of inflammatory control, reduction in peri-implant pocket depth, bleeding during probing, and reduction in bone loss in the dental implant region. Although, some literature is collected, some comparison in Table is not easy to understand. Therefore, I would suggest authors may revise or add some more Tables or Schemes before resubmission. Here are the comments and suggestions:

We appreciate all suggestions. Modifications made to the text were highlighted in red.

1. Please revise Table 1, it is not easy to read and compare each other.

Ok. Table 1 has been revised to incorporate necessary corrections. Additionally, improvements have been made to the discussion text. Special attention has been given to ensure that readers can effectively compare the results across different studies.

2.The conclusions should be extended and please add perspective.

Ok. The conclusion has been thoroughly reviewed and modified in accordance with the suggestions provided by the reviewer.

Reviewer 3 Report

Comments and Suggestions for Authors

The review assesses the influence of different photosensitizers on the outcomes of Photodynamic Therapy (PDT) for treating peri-implantitis. It examines the effectiveness of PDT in controlling inflammation, reducing peri-implant pocket depth, and limiting bone loss around dental implants. The study emphasizes the importance of selecting an appropriate photosensitizer based on patient-specific characteristics and peri-implantitis conditions, as it significantly affects the treatment's efficacy, impact on oral microbiome, and clinical patterns of peri-implantitis.

The review identified several limitations in the paper:

1. The study did not provide detailed information about the specificity of photosensitizers towards different bacterial species. This lack of specificity can lead to a broad impact on the oral microbiome, potentially affecting beneficial bacteria.

2. The paper did not address the long-term efficacy of PDT in managing peri-implantitis. This gap leaves questions about the sustainability of treatment outcomes and the potential for recurrence of the condition.

3. There was no direct comparison between PDT and traditional treatment methods for peri-implantitis. Such a comparison is crucial to establish the relative effectiveness, advantages, and potential drawbacks of PDT over standard care options.

4. The paper did not discuss the effects of photosensitizers and the PDT process on host tissues surrounding the implants. Understanding these impacts is essential for assessing the safety and potential side effects of the treatment.

5. The study lacked in-depth mechanistic insights into how different photosensitizers affect the treatment process. A thorough understanding of the mechanisms can guide the selection of photosensitizers and optimization of treatment protocols.

6. There was no detailed discussion on how patient-specific factors (like genetics, immune status, and oral hygiene) influence the outcomes of PDT. Such information is vital for personalized treatment planning and predicting treatment success.

Author Response

Reviewer's comments

Answers

The review assesses the influence of different photosensitizers on the outcomes of Photodynamic Therapy (PDT) for treating peri-implantitis. It examines the effectiveness of PDT in controlling inflammation, reducing peri-implant pocket depth, and limiting bone loss around dental implants. The study emphasizes the importance of selecting an appropriate photosensitizer based on patient-specific characteristics and peri-implantitis conditions, as it significantly affects the treatment's efficacy, impact on oral microbiome, and clinical patterns of peri-implantitis.

The review identified several limitations in the paper:

We appreciate all suggestions. Modifications made to the text were highlighted in red.

1. The study did not provide detailed information about the specificity of photosensitizers towards different bacterial species. This lack of specificity can lead to a broad impact on the oral microbiome, potentially affecting beneficial bacteria.

Ok. The requested information has been incorporated into the introduction as per the suggestion provided by the reviewer.

2. The paper did not address the long-term efficacy of PDT in managing peri-implantitis. This gap leaves questions about the sustainability of treatment outcomes and the potential for recurrence of the condition.

Follow-up information from all included studies has been collected and is presented in Table 1. Additionally, in the discussion, particularly in section 4.8, aspects related to monitoring are thoroughly discussed.

3. There was no direct comparison between PDT and traditional treatment methods for peri-implantitis. Such a comparison is crucial to establish the relative effectiveness, advantages, and potential drawbacks of PDT over standard care options.

Information regarding the comparison between the different strategies is discussed in the discussion section, particularly in items 4.5 and 4.7.

It is noteworthy that among the 7 studies included in the systematic review, only 1 did not implement manual or surgical therapy before the PDT treatment.

4. The paper did not discuss the effects of photosensitizers and the PDT process on host tissues surrounding the implants. Understanding these impacts is essential for assessing the safety and potential side effects of the treatment.

The photosensitizers employed in the included studies exhibit selectivity for Gram-negative bacteria, thereby avoiding changes in tissues neighboring the implant.

This information has been inserted into both the introduction and item 4.8 of the discussion.

5. The study lacked in-depth mechanistic insights into how different photosensitizers affect the treatment process. A thorough understanding of the mechanisms can guide the selection of photosensitizers and optimization of treatment protocols.

Ok. The requested information has been inserted into the introduction section.

6. There was no detailed discussion on how patient-specific factors (like genetics, immune status, and oral hygiene) influence the outcomes of PDT. Such information is vital for personalized treatment planning and predicting treatment success.

Unfortunately, the selected studies do not address these factors, as they chose to evaluate patients with controlled pathologies. We concur with the reviewer regarding the significance of this information. Therefore, we have included a discussion of this aspect in item 4.8 of the discussion section. Furthermore, the importance of a thorough analysis has been emphasized in the conclusion of the study.

Round 2

Reviewer 2 Report

Comments and Suggestions for Authors

Table 1 can be separated into several small Tables.

Author Response

Reviewer's comments

Answers

We appreciate all suggestions. Modifications made to the text were highlighted in red.

1. Table 1 can be separated into several small Tables.

Ok. Table 1 was edited according to the reviewer's suggestions.
